# Characterization of Running Intensity in Canadian Football Based on Tactical Position

**DOI:** 10.3390/s24082644

**Published:** 2024-04-21

**Authors:** Abdullah Zafar, Samuel Guay, Sophie-Andrée Vinet, Amélie Apinis-Deshaies, Raphaëlle Creniault, Géraldine Martens, François Prince, Louis De Beaumont

**Affiliations:** 1Department of Kinesiology, University of Waterloo, Waterloo, ON N2L 3G1, Canada; a35zafar@uwaterloo.ca; 2Centre de Recherche, Hôpital du Sacré-Cœur de Montréal, Montreal, QC H4J 1C5, Canada; samuel.guay@umontreal.ca (S.G.); sophie-andree.vinet@umontreal.ca (S.-A.V.); raphaelle.creniault@umontreal.ca (R.C.); geraldine.martens@uliege.be (G.M.); louis.de.beaumont@umontreal.ca (L.D.B.); 3Department of Surgery, Faculty of Medicine, University of Montreal, Montreal, QC H3T 1J4, Canada; 4Institut National du Sport du Québec, Montreal, QC H1V 3N7, Canada

**Keywords:** wearable sensors, training, game, workload, physical performance, gaussian mixture model

## Abstract

This study aimed to use a data-driven approach to identify individualized speed thresholds to characterize running demands and athlete workload during games and practices in skill and linemen football players. Data were recorded from wearable sensors over 28 sessions from 30 male Canadian varsity football athletes, resulting in a total of 287 performances analyzed, including 137 games and 150 practices, using a global positioning system. Speed zones were identified for each performance by fitting a 5-dimensional Gaussian mixture model (GMM) corresponding to 5 running intensity zones from minimal (zone 1) to maximal (zone 5). Skill players had significantly higher (*p* < 0.001) speed thresholds, percentage of time spent, and distance covered in maximal intensity zones compared to linemen. The distance covered in game settings was significantly higher (*p* < 0.001) compared to practices. This study highlighted the use of individualized speed thresholds to determine running intensity and athlete workloads for American and Canadian football athletes, as well as compare running performances between practice and game scenarios. This approach can be used to monitor physical workload in athletes with respect to their tactical positions during practices and games, and to ensure that athletes are adequately trained to meet in-game physical demands.

## 1. Introduction

North American football is a dynamic team sport consisting of short, high-intensity intervals of work interspersed with longer periods of rest [1]. The intensity of physical exertion in such team sports is frequently characterized by velocity or acceleration profiles from running performances, which can be used to quantify the physical demands required for competition [2,3,4]. Since understanding these demands is crucial for determining appropriate training and physical preparation protocols, the way running intensity is quantified becomes critical, especially as North American football involves significant variations in running performance according to tactical position [5]. Furthermore, a distinction should be made between running performance and running intensity [6] in that the former is simply a global indication of running speed while the latter signifies running speed in relation to an athlete’s capability in terms of the speeds produced during performance.

To distinguish levels of physical performance, the previous literature has categorized running speeds into discrete bins using different speed thresholds. In a recent paper [5], researchers examined spatio-temporal variables (distance, velocity) and high velocity, acceleration, and deceleration efforts to compare positional demands in NFL players during games. Their effort-based parameters only considered sprinting speeds, which were established with an absolute speed threshold, potentially over- or underestimating the amount of high-speed running performed by players based on their sprint speeds [7]. In another paper [8], authors established their own speed zones as “Lo” (0–10 km/h), “Med” (10–16 km/h), “Hi” (16–23 km/h), and “Sprint” (>23 km/h). This approach included being stationary and walking at quite low speeds and should not be considered as a physiological demand but rather as a recuperation period. Based on the previous literature [9,10,11], five speed zones were selected (0–1, 1–6, 6–12, 12–16, and >16 km/h) that corresponded to standing, walking, jogging, running, and sprinting [12]. The limit of the approach is that the literature referenced to determine the zones were based on other team sports (soccer, Aussie rules football, and rugby), and as such, did not consider the real distribution of speed for selecting zones that will account for player positions in American football. Additionally, Canadian varsity football follows a slightly different game structure and level of play than professional American football, and as such, the characterization of running demands may also differ and has not been explored.

Approaches to determining speed zones to date have been limited by a percentile-based approach which assumes that running effort speeds are evenly spaced [5]. Furthermore, percentile and percent of maximum velocity based approaches also suffer from the arbitrary selection of percent values to define thresholds as absolute speed threshold methods. By contrast, the data-driven approach proposed in this article is based on a Gaussian mixture model (GMM) which could refine our understanding of overall running performance by identifying clusters of running speeds within the overall running performance. Additionally, since the GMM can be used to find custom speed zones in individual performances, the resulting speed thresholds are truly reflective of running intensity rather than simply running performance [6]. As such, the approach can be used to understand key performance measures, like time spent or distance covered in high-intensity running [5,8], while naturally considering player position and fitness level.

The purpose of this study is therefore to use a data-driven approach to identify speed thresholds for running intensity in Canadian varsity football players, and to compare the physical demands of different tactical positions relative to their respective physical capacities.

## 2. Materials and Methods

### 2.1. Participants

A total of 30 male athletes were included in the study, with 19 skill players (age: 23.1 ± 1.6 years; height: 1.79 ± 0.06 m; weight: 85.6 ± 9.1 kg; 6 receivers, 10 defensive backs, 2 running backs, 1 linebacker) and 11 linemen (age: 22.9 ± 1.8 years; height: 1.88 ± 0.04 m; weight: 120.4 ± 9.6 kg; 6 offensive linemen, 5 defensive linemen). All participants met the eligibility criteria of being Canadian varsity football athletes who played in the 2021 season, and provided written informed consent. The study was reviewed and received ethics clearance through the Aging-Neuroimaging Research Ethics Committee of the CIUSSS Centre-Sud-de-l’Île-de-Montréal Office of Research Ethics Committee (ethics no. MP-53-2020-191).

### 2.2. Procedures

Data were collected over 28 sessions (16 games, 12 practices), for a collective total of 311 performances recorded (156 games, 155 practices). Sessions where athletes did not complete the game or practice, either due to injury or tactical reasons, were excluded from the analysis, resulting in 287 performances analyzed (137 games, 150 practices). Athletes were assigned the same Catapult sensor (Vector S7, Catapult Sports, Melbourne, Australia) for each session, which was embedded in tightly fitted vests (Figure 1) and collected activity profiles from a global positioning system (GPS) and heart rate sensors at 10 Hz, as well as inertial measurement units (IMU) at 100 Hz [13].

To identify speed zones, clusters of common running speeds were identified using a GMM. Based on the concept that a probabilistic distribution of running speeds from an athlete’s performance can be decomposed into a finite number of Gaussian distributions, in this context, each component Gaussian would represent a cluster of speeds achieved by the athlete (a speed “zone”). To fit the GMM, speeds below 5 km/h were first excluded from the data to remove the effect of standing or slow walking dominating the detection of speed clusters. The residual velocity data from the Catapult GPS units were binned (bin widths of 0.5 km/h) and 1 to 10 dimensional GMMs were fit using the Gaussian mixture function from scikit-learn [14]. The corresponding Bayesian information criterion (BIC) for each of the 1 to 10 dimensional models were also computed to determine an appropriate level of model complexity (i.e., how many zones to model). A dimensionality of 5 was selected as higher dimensionality did not improve the BIC, and 5 zones generally corresponds to previously identified speed zones [12], thereby translating to interpretable zones: minimal-intensity standing/walking (zone 1), low-intensity running (zone 2), medium-intensity running (zone 3), high-intensity running (zone 4), and maximal-intensity sprinting (zone 5). An example of the speed distribution for a receiver and offensive linemen are shown in Figure 2, along with the corresponding GMM.

### 2.3. Statistical Analysis

The thresholds separating speed zones were computed as the average of the mean plus or minus the standard deviation of the Gaussian distributions for two adjacent zones. The threshold values separating each zone were statistically compared using a Kruskal–Wallis test to determine the effect of athlete position on speed thresholds. Once zones were established, the start and end of each session was determined by the first and last entry into zone 4 (high-intensity running) and these were used to determine the session duration.

Using the threshold values, the time spent in each speed zone was computed by counting the number of samples where velocity was within the speed range of a zone and dividing by the sampling frequency; a percentage of time spent in each zone was then obtained by dividing the absolute time spent by the session duration. The percentage of time spent in each zone was compared using a Kruskal–Wallis test to determine the effect of athlete position. Using the identified samples, the distance covered in each speed zone was also computed by summing the velocities (in m/s) for those samples and dividing by the sampling frequency. The distance covered was similarly compared across positions for each speed zone using a Kruskal–Wallis test.

To compare running intensity across performance context, the speed thresholds, percentage of time spent in each zone, and distance covered in each zone were averaged for every athlete based on performance context (game or practice). The average values were then compared between contexts for each speed zone using a Wilcoxon rank-sum test for each athlete group (skill players or linemen).

To explore the differences between extreme and average sprinting demands in-game, the peak intensity period for each game performance was identified as the 5 min period with the largest percentage of time spent in zone 5 (i.e., sprinting). The average sprinting demands were computed from the rest of the session, excluding the peak intensity period as well as any prolonged periods of only low-intensity running or standing, where no sprints occurred for at least 5 min. Sprint efforts were identified as entries into zone 5 which lasted for at least 1 s. The number of sprint efforts per minute, the average time of recovery between sprints, and the average distance covered in a sprint were compared between peak intensity and average periods using a Wilcoxon rank-sum test for each athlete group (skill players or linemen).

All statistical comparisons were performed in R [15] with significance levels for all tests set at *p* = 0.05, and Dunn’s test with Holm–Bonferroni correction was used as a post hoc test for pairwise comparisons for the Kruskal–Wallis test. Wilcoxon effect sizes were calculated with values of 0.3, 0.5, and 0.7 being threshold values for small, medium, and large effect sizes, respectively. The median and interquartile range were used as summary statistics.

## 3. Results

Speed threshold values were significantly different across athlete positions (Figure 3). In particular, the Kruskal–Wallis test showed significant differences across positions for speed thresholds between zones 3–4 (H(5) = 134, *p* < 0.001) and 4–5 (H(5) = 135, *p* < 0.001). Pairwise comparisons between positions showed that receivers and defensive backs had higher threshold speeds for zones 3/4/5 compared to all linemen (*p* < 0.01). Linebackers and running backs only had higher speed threshold compared to offensive linemen (*p* < 0.05).

In general, the speed thresholds for skill players were higher than linemen, with maximal intensity running thresholds of 18.9 km/h (IQR: 2.6 km/h) or 5.2 m/s (IQR: 0.7 m/s) for skill players and 14.3 km/h (IQR: 3.9 km/h) or 4.0 m/s (IQR: 1.1 m/s) for linemen. Receivers, defensive backs, and linebackers had the highest thresholds (18.9–19.1 km/h or 5.2–5.3 m/s) while offensive linemen had the lowest thresholds (12.8 km/h or 3.6 m/s).

The percentage of time spent, and the distance covered in each speed zone were also significantly different across athlete positions (Figure 4) for all zones. Receivers and defensive backs spent less percentage of time in zone 1 (minimal intensity) than all other positions (*p* < 0.01), while also covering more distance compared to running backs and linemen (*p* < 0.05). In all other speed zones, receivers and defensive backs spent more percentage of time (*p* < 0.01) and covered more distance (*p* < 0.01) compared to running backs and linemen. Receivers spent more percentage of time (*p* = 0.04) in zone 5 (maximal intensity) compared to defensive backs, although there was no significant difference in distance covered between the two. Within their respective speed zones, skill players (primarily receivers and defensive backs) covered more distance (934 m, IQR: 318 m) and spent a greater percentage of time (2.1%, IQR: 0.6%) in high-maximal intensity zones compared to linemen (435 m, IQR: 219 m; 1.4%, IQR: 0.8%).

Comparing game and practice contexts, the overall duration of game sessions (190 min, IQR: 35 min) was greater than practice sessions (123 min, IQR: 14 min) and demonstrated a large effect size (*p* < 0.001, r = 0.68). Wilcoxon rank-sum tests showed that for skill players, detected speed thresholds were significantly lower in practice contexts compared to game contexts only for the threshold between zones 2–3 (*p* < 0.01) and the effect size for this difference was moderate (r = 0.44). Furthermore, the percentage of time spent in all zones for skill players showed no significant difference across contexts, although the distance covered in all speed zones was significantly higher in game contexts (*p* < 0.001) with moderate-to-large effect sizes (0.36 ≤ r ≤ 0.55). For linemen, there were no significant differences in speed thresholds between game and practice contexts. The percentage of time spent in all speed zones also showed no significant difference between game and practice contexts for linemen, while the distance covered in all zones were also significantly higher in game contexts (*p* < 0.001) with large effect sizes (0.50 ≤ r ≤ 0.84). Speed thresholds for game and practice contexts are shown in Figure 5, and percentage of time spent and distance covered in each zone are shown in Figure 6.

The number of sprints per minute, the average recovery time between sprints, and the average distance between sprints (Figure 7) were all significantly different between the peak intensity and average running performance (*p* < 0.001), with moderate-to-large effect sizes (0.40 ≤ r ≤ 0.71). For skill players, the sprint rate of 0.35 (IQR: 0.10) sprints per minute on average increased to 0.80 (IQR: 0.4) sprints per minute, while for linemen, the average performance sprint rate of 0.30 (IQR: 0.14) increased to 0.60 (IQR: 0.40) sprints per minute. Both athlete groups had shorter recovery times between sprints in the peak intensity period compared to average, with skill players reducing recovery time from 90 s (IQR: 31 s) to 58 s (IQR: 44 s) and linemen reducing recovery time from 84 s (IQR: 55 s) to 39 s (IQR: 56 s). Finally, skill players increased the average sprint distance covered in the peak intensity period from 33 m (IQR: 5 m) to 39 m (IQR: 12 m), while linemen had an increase from 28 m (IQR: 10 m) to 32 m (IQR: 15 m).

## 4. Discussion

The heterogeneity in running speed distributions across tactical positions, as evidenced by the variance in detected speed thresholds, illustrates the need for individualized thresholds for workload monitoring in American and Canadian football athletes. While previously, only maximum velocities reached were reported across tactical positions [5] (Sanchez et al., 2023), a similar result was found that offensive linemen had the lowest velocity output, and the highest velocities were achieved by receivers and defensive backs. Importantly, using thresholds being detected via GMM for each session independently, a unified set of running performance was detected across all athletes and tactical positions. As indicated from Figure 2, the natural distribution of running speeds follows a multi-modal trend with running performed at several clusters of speed values rather than a uni-modal trend. Due to this multi-modal nature, percentile and percent of maximum running velocity based approaches generate threshold values that divide a single speed cluster unintuitively (e.g., the threshold between Zone #1 and #2 for the offensive lineman in Figure 2). Furthermore, the selection of percentage values to define percentiles defining speed zone ranges can become as arbitrary as defining absolute speed thresholds, as can been noted with the varying use of 80th, 85th, or 90th percentile for sprinting [16] (Buchheit et al., 2021). Additionally, the use of individualized running thresholds in soccer have demonstrated reduced inter-match variability compared to generic speed thresholds when evaluating high-speed running performances [17] (Carling et al., 2016), indicating an increased robustness of such an approach in workload monitoring. The similarity in detected thresholds for groups of playing positions (e.g., receivers, defensive backs, and linebackers) may also indicate larger groupings of tactical positions in terms of physical preparation beyond distinctions of skill players versus linemen, or offensive versus defensive players. Future work may benefit from exploring the heterogeneity in the characterization of physical profiles in American football and extracting clusters of players, which may inform the grouping of athletes during training or scouting.

When comparing running intensities across game and practice contexts, the distances covered were much higher in game settings, which may be attributed to the longer duration of games compared to practices. By contrast, the percentages of time in each zone were nearly identical across contexts, highlighting a consistency in the composition of running performances for each position. Additionally, the detected threshold values using the GMM method were generally consistent across game and practice contexts, despite the large discrepancies in athlete play time across contexts. While previous studies in American football have only looked at practice and game contexts separately, there was less discrepancy between the distances covered in practice [12] (DeMartini et al., 2011) and games [5] (Sanchez et al., 2023). This is particularly striking as the structure of games and practices may differ quite dramatically, with practices not only simulating games (e.g., scrimmages) but also consisting of a variety of strength and conditioning drills. However, discrepancies between game and practice intensity in both absolute demands (such as distance covered) and relative demands (such as percentage of time) should be considered to improve physical preparation to reach target demands during games.

Furthermore, the quantification of the running performance during peak-intensity periods during game settings indicates the peak physical performance capacity required of athletes. Notably, the average play clock (i.e., time allowed to elapse between the end of one play and the start of another) in Canadian football is 25 s, which in peak-intensity periods was close to the average recovery time between sprints for linemen, while for skill players the average recovery time between sprints was closer to a minute. In American football, rest times between plays have been quantified as about 29.6–36 s [18] (Plisk and Gambetta, 1997). As such, even in peak intensity periods of a game, skill players do not perform maximal intensity running in every play, while linemen do perform maximal intensity efforts every play during peak intensity periods. While peak intensity periods in team sports typically underestimate true physical capacity in athletes, the characterization of high-intensity work performed during these periods may inform the evaluation of work completed in simulated game scenarios during practice [19] (Weaving et al., 2022). In practice, the exposure of athletes to bouts of running relative to their own maximum velocity has shown a reduction in injury in Gaelic football [20] (Malone et al., 2017), indicating the practical application of the characterization of the maximal intensity running in peak intensity periods presented here towards the design of physical preparation protocols.

The present study on the application of GMMs to identify speed thresholds in football players is limited by the small sample size in certain athlete positions as well as the inclusion of only two teams over the course of a single season. As such, the lower threshold values and distances covered for running backs may be due to the characteristics of the single athlete collected in that position rather than reflective of the positional demands in general. Further exploration including more athletes (running backs and linebackers in particular), and across more seasons is warranted. The consideration of tactical factors such as game context in terms of scores was also not factored into the analysis of thresholds, although it may have an impact on the relative playing time and actions of offensive and defensive players. While in general the playing time may not influence the threshold values detected by GMMs, as demonstrated by the similarity in thresholds across practice and game contexts, the actions and plays run by the athletes may be affected by the scoreline, thus changing the overall running performance and threshold values. Furthermore, the de-composition of practice sessions into individual positions or team-based drills, as well as physical conditioning drills, may be of interest to explore relative running intensities across practice and game components. Finally, the current study did not consider the overall team performance of the two teams involved in terms of team ranking, which would be valuable in assessing the qualities of athletes in high- versus low-performing teams.

## 5. Conclusions

The findings from this study demonstrate the utility of a data-driven approach using wearable sensors to identify speed thresholds in American and Canadian football to appropriately define running intensity demands according to individual player capacities. Using such an approach, high-intensity running for linemen was characterized as medium-intensity running for skill players, thus highlighting the need to distinguish physical outputs (in terms of pure speed) from physical demands (in terms of intensity). The ability to characterize the intensity of physical workload performed relative to each athlete’s capacity rather than an absolute threshold may be of use when evaluating physiological responses, such as heart rate, to gauge physical fatigue. Reporting of time spent and distances covered according to running intensity zones can provide insight into the workload during a session and more accurately reflect the effects of fatigue experienced by an athlete. Furthermore, this is the first study to use Gaussian mixture models to determine speed thresholds in Canadian football, although the approach has also been used in other team sports for speed zone identification [21] (Park et al., 2019). Further investigation of patterns of fatigue in American football athletes would benefit from relating the running intensity zones identified in this work to additional physiological measures, such as heart rate.

## Figures and Tables

**Figure 1 sensors-24-02644-f001:**
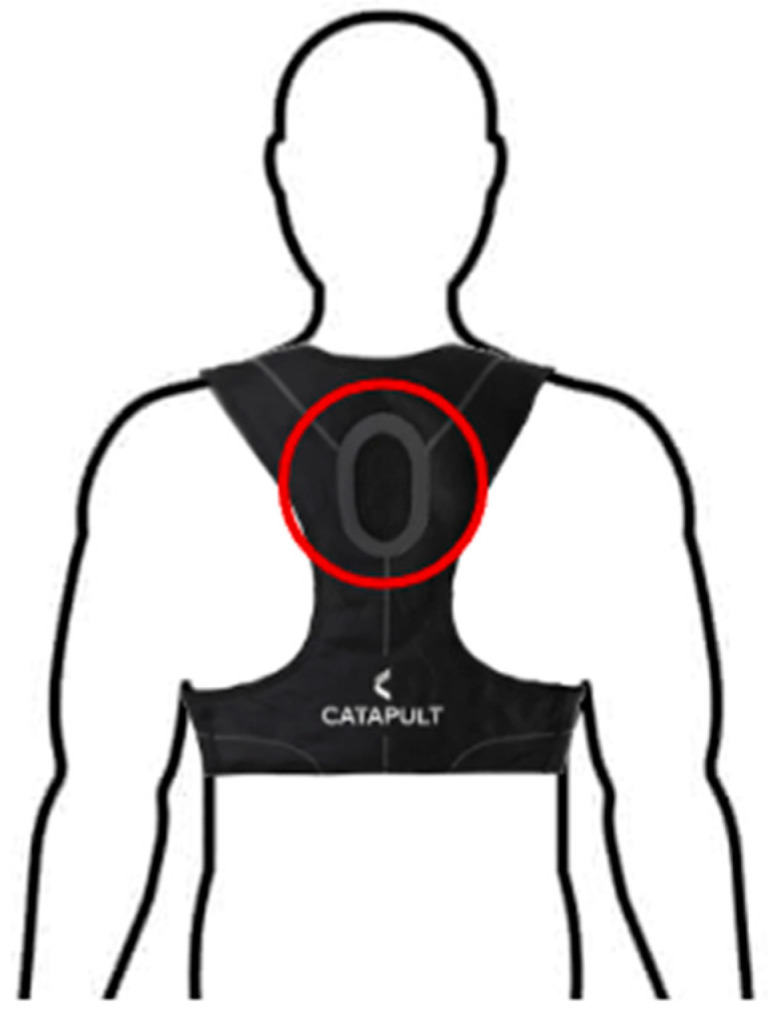
Placement of the Catapult sensor. Each athlete wore a fitted vest with a sleeve housing the catapult sensor on the upper torso (circled in red).

**Figure 2 sensors-24-02644-f002:**
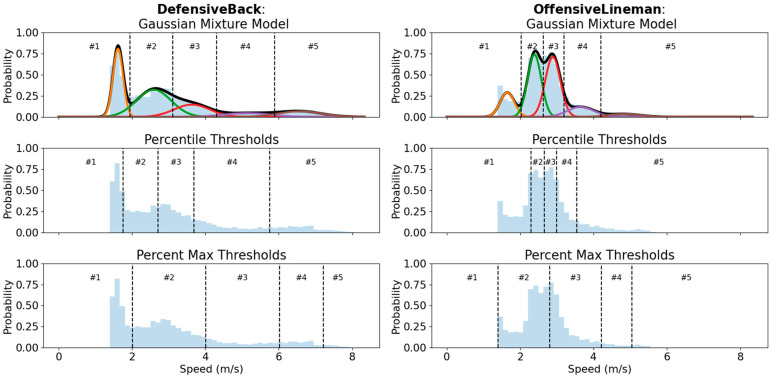
Gaussian mixture model speed zone detection. Distribution of running speeds during a session for a receiver (**left**) and offensive lineman (**right**) with the corresponding Gaussian distributions and speed thresholds overlaid for each zone: #1 (orange), #2 (green), #3 (red), #4 (purple), #5 (brown), and overall GMM (black). For comparison, percentile-based thresholds are shown for 25th, 50th, 75th, and 90th percentile values separating zones, and percent max-based thresholds are shown for 25%, 50%, 75%, and 90% of maximum velocity value separating zones.

**Figure 3 sensors-24-02644-f003:**
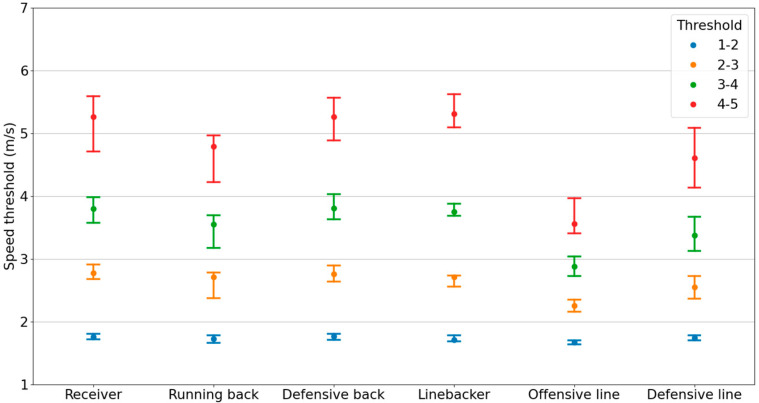
Speed zone thresholds by athlete position. The median and interquartile range of the speed thresholds for each speed zone, separated by athlete position.

**Figure 4 sensors-24-02644-f004:**
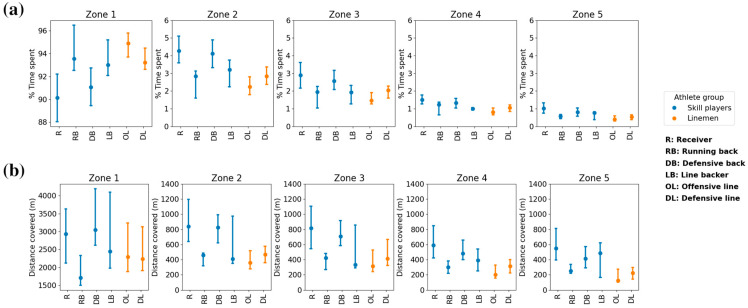
Percentage of time spent, and distance covered in each speed zone. Median and interquartile range of the percentage of time spent (**a**) and distance covered (**b**) in each speed zone are plotted separately for each tactical position and colored by athlete type.

**Figure 5 sensors-24-02644-f005:**
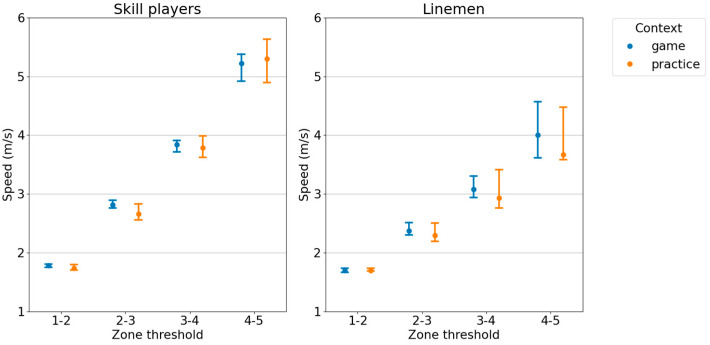
Game and practice speed thresholds. The median and interquartile range of detected speed thresholds for game and practice settings, split by athlete type.

**Figure 6 sensors-24-02644-f006:**
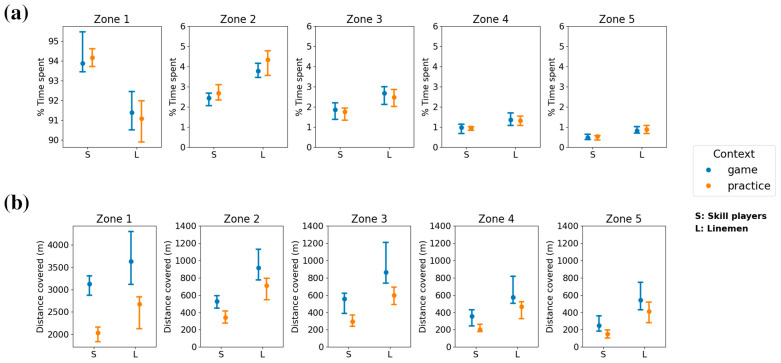
Percentage of time spent, and distance covered in each speed zone based on context. The median and interquartile range of the percentage of time spent (**a**) and distance covered (**b**) in each speed zone for game and practice settings, split by athlete type.

**Figure 7 sensors-24-02644-f007:**
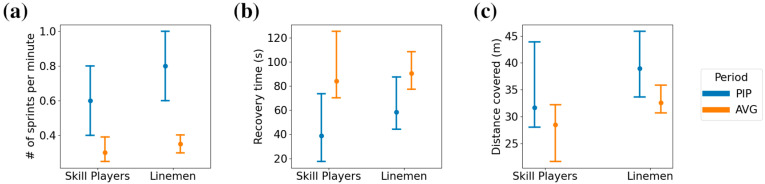
Characterization of peak intensity periods in game settings. The median and interquartile range of (**a**) the number of maximal intensity efforts performed per minute, (**b**) the average recovery time between maximal intensity efforts, and (**c**) the average distance covered per maximal intensity effort during the peak intensity period (PIP) and the match average (AVG).

## Data Availability

Data sharing is not applicable.

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
