# Peer review of "Characterization of Running Intensity in Canadian Football Based on Tactical Position"

_sensors, 2024, doi:10.3390/s24082644_

Round 1

Reviewer 1 Report

Comments and Suggestions for Authors

This study aimed to use a data-driven approach (GMM) to identify speed thresholds for running demands in Canadian football. My primary criticism is the authors wish to express superiority of the GMM method, but there is no comparison to the currently accepted method. I have detailed other minor criticisms below, but this is worth stating outright.

Abstract:

1. Consider adding data if there is room

Introduction

1. Lns 38-40: I'm confused by this. What "capacity" values were used in GMM? This method still seems subjected to the speed the athlete produced. How is the GMM method different than relative running speeds calculated from maximal sprint speed?

2. Lns 60-69: need more information to help the reader differentiate between GMM and the percentile-based approach. Is there a figure that could be created to help with this? 

Methods:

1. Include participant eligibility criteria.

2. Was the same GPS unit used for each athlete each day?

3. What trim and split procedures were employed, if any?

4. Figure 1: Please label the y-axis

5. Statistical analysis: why were non-parametric analyses used for continuous variables? Were the variables not normally distributed?

6. Statistical analysis: add details of the effect sizes calculated and thresholds

Discussion:

1. Ln 238: I don't think your data support "the power of the data-driven approach" because there is no comparison to the percentile approach. Please consider including this comparison.

2. Ln 240: provide some detail regarding the "greater stability" suggested from the soccer study

3. Were there any considerations of athlete play time? How would time on the field affect these data?

4. Lns 282-293: what is going on here? These appear to be sentences that are not linked or well-explained. 

5. Please include a paragraph on limitations of the study

Conclusions:

1. I don't think there is a solid argument outlining the differences between GMM and the percentile-based model. This simply presents a new method and does not exercise superiority over another method. 

Author Response

REVIEWER #1:

We would like to thank this reviewer for all the comments provided.

This study aimed to use a data-driven approach (GMM) to identify speed thresholds for running demands in Canadian football. My primary criticism is the authors wish to express superiority of the GMM method, but there is no comparison to the currently accepted method. I have detailed other minor criticisms below, but this is worth stating outright.

Abstract:

  1. Consider adding data if there is room

Thank you for the comment, adding data into the abstract was considered however omitted for space constraints.

Introduction

  1. Lines 38-40: I'm confused by this. What "capacity" values were used in GMM? This method still seems subjected to the speed the athlete produced. How is the GMM method different than relative running speeds calculated from maximal sprint speed?

Thank you for your comment, “capacity” here is relative to speeds produced by the athletes and this has been clarified on Lines 38-40 with the following addition: “...the latter signifies running speed in relation to an athlete’s capability in terms of the speeds produced during performance.”

However, the difference is using GMMs compared to relative speeds (i.e. %max speed) is the distribution of running speeds that is not uniform and so choosing % based thresholds can become as arbitrary as choosing absolute velocity thresholds. This is now indicated in the introduction as well: “Furthermore, percentile and percent of maximum velocity based approaches also suffer from the arbitrary selection of percent values to define thresholds as absolute speed threshold methods.“

As shown in the additions to Figure 2, relative speed thresholds based on percentile or %max approaches produce unintuitive boundaries between speed zones, whereas the GMM takes the multi-modal nature of the running performance into account to estimate more natural boundaries in the data.

  1. Lines 60-69: need more information to help the reader differentiate between GMM and the percentile-based approach. Is there a figure that could be created to help with this?

Additions were made to Figure 2 to demonstrate these differences.

Methods:

  1. Include participant eligibility criteria.

Additions were made to the text to indicate the eligibility criteria: “All participants met eligibility criteria of being Canadian varsity football athletes who played in the 2021 season and were provided written informed consent.”

  1. Was the same GPS unit used for each athlete each day?

Yes, the same catapult unit was used for each athlete each day - text was added to clarify this in the methods: “Athletes were assigned the same Catapult sensor for each session which was embedded in tightly fitted vests, which collected activity profiles from a global positioning system (GPS) and heart rate sensor at 10 Hz, as well as inertial measurement units (IMU) at 100 Hz.”

  1. What trim and split procedures were employed, if any?

Thank you for your comment, If by time/split procedures you are referring to train/test splits for the GMM then there were none applied as it was being used purely as an unsupervised approach to find clusters in the running performances for extracting speed thresholds, and not for prediction or classification in a supervised manner.

  1. Figure 1: Please label the y-axis

Thank you, the y-axis is now labeled. Figure 1 had now also been re-numbered as Figure 2, due to the addition of the figure indicating sensor placement.

  1. Statistical analysis: why were non-parametric analyses used for continuous variables? Were the variables not normally distributed?

Yes, since the data didn’t pass assumptions of normality and homogeneity of variances which is why non-parametric tests were used instead.

  1. Statistical analysis: add details of the effect sizes calculated and thresholds

Thank you, these details were added in the methods: “Wilcoxon effect sizes were calculated with values of 0.3, 0.5, and 0.7 being threshold values for small, medium, and large effect sizes, respectively.”

Discussion:

  1. Line 238: I don't think your data support "the power of the data-driven approach" because there is no comparison to the percentile approach. Please consider including this comparison.

Thank you for your comment, the additions to Figure 2 help to demonstrate the limitations of percentile/percent-max approaches. Additionally, the following lines were added to the discussion to help elaborate: “As indicated from Figure 2, the natural distribution of running speeds follows a multimodal trend with running performed at several clusters of speed values rather than a unimodal trend. Due to this multi-modal nature, percentile and percent of maximum running velocity based approaches generate threshold values which divide a single speed cluster unintuitively (e.g. the threshold between Zone #1 and #2 for the offensive lineman in Figure 2). Furthermore, the selection of percentage values to define percentiles defining speed zone ranges can become as arbitrary as defining absolute speed thresholds, as can be noted with the varying use of 80th, 85th, or 90th percentile for sprinting (Buchheit et al., 2021).”

  1. Line 240: provide some detail regarding the "greater stability" suggested from the soccer study

            The sentence has been modified to give more detail about the manner of the stability: “Additionally, the use of individualized running thresholds in soccer has demonstrated reduced inter-match variability compared to generic speed thresholds when evaluating high-speed running performances (Carling et al., 2016), indicating an increased robustness of such an approach in workload monitoring.“

  1. Were there any considerations of athlete play time? How would time on the field affect these data?

There were considerations of athlete play times when comparing practice and game contexts, which had significantly different session times (game: 190 min, IQR 35 min; practice: 123 min, IQR 14 min). Despite the differences in play time, the speed thresholds detected were largely consistent between game and practice sessions. The discussion now alludes to this: “Additionally, the detected threshold values using the GMM method were generally consistent across game and practice contexts, despite the large discrepancies in athletes play time across contexts.”

  1. Lines 282-293: what is going on here? These appear to be sentences that are not linked or well-explained.

Thank you, these sentences have been removed and replaced with a paragraph on limitations.

  1. Please include a paragraph on the limitations of the study

Thank you, the following paragraph was added to the discussion: “The present study on the application of GMMs to identify speed thresholds in football players is limited by the small sample size in certain athlete positions as well as the inclusion of only two teams over the course of a single season. As such, the lower threshold values and distances covered for running backs may be due to the characteristics of the single athlete collected in that position rather than reflective of the positional demands in general. Further exploration, including more athletes (running backs and linebackers in particular), and across more seasons is warranted. The consideration of tactical factors such as game context in terms of scores was also not factored into the analysis of thresholds, although it may have an impact on the relative playing time and actions of offensive and defensive players. While in general the playing time may not influence the threshold values detected by GMMs, as demonstrated by the similarity in thresholds across practice and game contexts, the actions and plays run by the athletes may be affected by the scoreline, thus changing the overall running performance and threshold values. Furthermore, the decomposition of practice sessions into individual position or team-based drills, as well as physical conditioning drills, may be of interest to explore relative running intensities across practice and game components. Finally, the current study did not consider the overall team performance of the two teams involved in terms of team ranking which would be valuable in assessing the qualities of athletes in high versus low-performing teams.“

Conclusions:

  1. I don't think there is a solid argument outlining the differences between GMM and the percentile-based model. This simply presents a new method and does not exercise superiority over another method.

Thank you for your comment, while the study did not aim to replace absolute speed, percentile, or percent max-based approaches, amendments were made to Figure 2 and the introduction to highlight the advantages of GMMs over these other methods.   

Reviewer 2 Report

Comments and Suggestions for Authors

This intriguing manuscript on identifying speed thresholds is well-written and conveys a research effort that appears to be a component of an important trend in sports to better understand player performance through the use of wearable technology.  Overall, the paper is very clear, and seems to validate similar works in the literature. More vivid added value on the research may be obtained through several suggestions for the authors to entertain. 

1.      The abstract suggests the paper will offer some hope about gaining insight into player fatigue, but the manuscript does not return to this issue.  The declaration of thresholds has the potential to inform other efforts to categorize player performance, but the manuscript is descriptive in this regard but does not link thresholds to performance.  It is not clear if the homogeneity of the results precluded that step. 

2.      The work successfully has applied the Gaussian Mixture model to the collected dataset and there appears to be a precedent for this strategy.  The paper mentions the use of the model but it is unclear if testing the model provided additional insight on what in principle could have been established by another analytic method.

3.      The exploration of fatigue is limited characterization of speed and distance traveled, but there is no other data presented and integrated together to further inform a coaching or training staff.  Alone this is not critical path to utility of the data, but were there other dataset that are under development of that have been considered for a more comprehensive insight on fatigue? The question arises whether this dataset could be used to improve player performance or reduce risk of injury.  Either way, the reader could benefit from more insight on actual or proposed uses of the data.

4.      Figures 3 and 4 contain an abundance of information, but it remains unclear how the reader is expected to interpret the number of very small plots. Many statistics are presented and discussed, but the figures are not posited to be informative “as-is.”

5.      The manuscript does an excellent job with statistical analysis and description of the extensive data collected and the use of the GMM to partition datasets, though it seems the insight gained from the effort is understated.

Author Response

REVIEWER #2:

We would like to thank this reviewer for the comments provided.

This intriguing manuscript on identifying speed thresholds is well-written and conveys a research effort that appears to be a component of an important trend in sports to better understand player performance through the use of wearable technology.  Overall, the paper is very clear, and seems to validate similar works in the literature. More vivid added value on the research may be obtained through several suggestions for the authors to entertain.

  1. The abstract suggests the paper will offer some hope about gaining insight into player fatigue, but the manuscript does not return to this issue. The declaration of thresholds has the potential to inform other efforts to categorize player performance, but the manuscript is descriptive in this regard but does not link thresholds to performance.  It is not clear if the homogeneity of the results precluded that step.

            Thank you for your comment. The phrasing “monitor physical fatigue” has been replaced with “monitor physical workload” in the abstract.

  1. The work successfully has applied the Gaussian Mixture model to the collected dataset and there appears to be a precedent for this strategy. The paper mentions the use of the model but it is unclear if testing the model provided additional insight on what in principle could have been established by another analytic method.

Thank you for the comment, additions were made to Figure 2 comparing the GMM approach to percentile approaches. The difference in using GMMs compared to relative speeds (i.e. %max speed) is the distribution of running speeds that is not uniform and so choosing % based thresholds can become as arbitrary as choosing absolute velocity thresholds. This is now indicated in the introduction: “Furthermore, percentile and percent of maximum velocity based approaches also suffer from the arbitrary selection of percent values to define thresholds as absolute speed threshold methods.“

Further elaboration on the comparison between GMM and percent based approaches had also been added to the discussion: “As indicated from Figure 2, the natural distribution of running speeds follows a multimodal trend with running performed at several clusters of speed values rather than a unimodal trend. Due to this multi-modal nature, percentile and percent of maximum running velocity based approaches generate threshold values which divide a single speed cluster unintuitively (e.g., the threshold between Zone #1 and #2 for the offensive lineman in Figure 2). Furthermore, the selection of percentage values to define percentiles defining speed zone ranges can become as arbitrary as defining absolute speed thresholds, as can be noted with the varying use of 80th, 85th, or 90th percentile for sprinting (Buchheit et al., 2021).”

  1. The exploration of fatigue is limited characterization of speed and distance travelled, but there is no other data presented and integrated together to further inform a coaching or training staff. Alone this is not critical path to utility of the data, but were there other dataset that are under development of that have been considered for a more comprehensive insight on fatigue? The question arises whether this dataset could be used to improve player performance or reduce risk of injury.  Either way, the reader could benefit from more insight on actual or proposed uses of the data.

Thank you for the comment. Yes, this method will be used in future work to better quantify fatigue in conjunction with heart rate data by a more accurate characterization of physical effort exerted based on relative athlete capacity. The following has now been alluded to in the conclusion section: “The ability to characterize the intensity of physical workload performed relative to each athlete’s capacity rather than an absolute threshold may be of use when evaluating physiological responses, such as heart rate, to gauge physical fatigue.”

  1. Figures 3 and 4 contain an abundance of information, but it remains unclear how the reader is expected to interpret the number of very small plots. Many statistics are presented and discussed, but the figures are not posited to be informative “as-is.”

Thank you, we agree with the abundance of information presented in these Figures (now Figures 4 & 5) but feel that they are all relevant in demonstrating the overall trends in running performances across playing positions and speed zones which are better presented as a figure compared to a data table.

  1. The manuscript does an excellent job with statistical analysis and description of the extensive data collected and the use of the GMM to partition datasets, though it seems the insight gained from the effort is understated.

Thank you, the added points in the introduction and discussion, as well as Figure 2, mentioned above now aim to distinguish the GMM approach from other analytical approaches, the utility of which is now alluded to in workload monitoring (in the discussion) and in evaluation of physical fatigue as future work (in the conclusion).

Reviewer 3 Report

Comments and Suggestions for Authors

Dear authors,

The article presents an innovative analysis regarding the characterization of American football athletes using GPS data during a significant number of practice sessions.

I have no major comments regarding the methodology and the results presented by the authors. However, there are aspects of the article presentation that must be addressed before publication:

  • In the introduction, the paragraphs appear to be correctly ordered without an extra line break, as seen in other sections. Please standardize the entire text to match the format in the introduction.

  • In line 90, when discussing the Catapult sensors, there is no mention of their technical characteristics, the company, or the location of their headquarters. This information is important for readers to have the necessary details to replicate your results.

  • I would appreciate if you could provide information regarding the location of the sensor on the athletes graphically, incorporating a diagram, image, or photograph of its placement. This can help in understanding the arrangement of the mentioned system in your methodology.

Author Response

REVIEWER #3:

We would like to thank this reviewer for the comments provided.

The article presents an innovative analysis regarding the characterization of American football athletes using GPS data during a significant number of practice sessions.

I have no major comments regarding the methodology and the results presented by the authors. However, there are aspects of the article presentation that must be addressed before publication:

In the introduction, the paragraphs appear to be correctly ordered without an extra line break, as seen in other sections. Please standardize the entire text to match the format in the introduction.

Thank you for your comment, the formatting has now been fixed.

In line 90, when discussing the Catapult sensors, there is no mention of their technical characteristics, the company, or the location of their headquarters. This information is important for readers to have the necessary details to replicate your results.

Thank you, the device, company and headquarters have now been added in the methods: “Athletes were assigned the same Catapult sensor (Vector S7, Catapult Sports, Melbourne, Australia) for each session which was embedded in tightly fitted vests (Figure 1), which collected activity profiles from a global positioning system (GPS) and heart rate sensor at 10 Hz, as well as inertial measurement units (IMU) at 100 Hz.”

I would appreciate if you could provide information regarding the location of the sensor on the athletes graphically, incorporating a diagram, image, or photograph of its placement. This can help in understanding the arrangement of the mentioned system in your methodology.

Thank you, a graphic has been added as Figure 1 demonstrating the location of the sensor and vest worn by the athletes.

Round 2

Reviewer 1 Report

Comments and Suggestions for Authors

My primary criticism remains that the authors wish to express superiority of the GMM method, but there is no comparison to the currently accepted method. If you aren't going to run comparison analysis, then remove the language that suggests GMM is better than the current models. Your data do not support any "better than" statements.

Author Response

Dear reviewer,

We have modified the manuscript at Line 244 accordingly to your comment.

It was our omission.

Reviewer 2 Report

Comments and Suggestions for Authors

Thank you for the revised paper that in current form reponds adequately to the prior review.  The effort to feature outcomes more clearly will benefit the journal readership.

Author Response

Thank you for your contribution to improve this manuscript